# Discovering Themes in Deep Brain Stimulation Research Using Explainable Artificial Intelligence

**DOI:** 10.3390/biomedicines11030771

**Published:** 2023-03-03

**Authors:** Ben Allen

**Affiliations:** Department of Psychology, University of Kansas, Lawrence, KS 66045, USA; benallen@ku.edu

**Keywords:** deep brain stimulation, explainable artificial intelligence, machine learning

## Abstract

Deep brain stimulation is a treatment that controls symptoms by changing brain activity. The complexity of how to best treat brain dysfunction with deep brain stimulation has spawned research into artificial intelligence approaches. Machine learning is a subset of artificial intelligence that uses computers to learn patterns in data and has many healthcare applications, such as an aid in diagnosis, personalized medicine, and clinical decision support. Yet, how machine learning models make decisions is often opaque. The spirit of explainable artificial intelligence is to use machine learning models that produce interpretable solutions. Here, we use topic modeling to synthesize recent literature on explainable artificial intelligence approaches to extracting domain knowledge from machine learning models relevant to deep brain stimulation. The results show that patient classification (i.e., diagnostic models, precision medicine) is the most common problem in deep brain stimulation studies that employ explainable artificial intelligence. Other topics concern attempts to optimize stimulation strategies and the importance of explainable methods. Overall, this review supports the potential for artificial intelligence to revolutionize deep brain stimulation by personalizing stimulation protocols and adapting stimulation in real time.

## 1. Introduction

Deep brain stimulation is a technique wherein an implanted device delivers an electrical current into a patient’s brain to improve their clinical condition [1]. Deep brain stimulation is an important therapy because it is relatively safe and applicable to a variety of neurological and psychiatric disorders [2]. While the underlying mechanisms are still under study, chronic electrical stimulation causes cellular and molecular changes in brain circuits that underlie behavior [2]. A crucial question in deep brain stimulation research is how to adapt neurostimulation to the patient. Artificial intelligence approaches offer solutions to this problem, such as altering stimulation schedules based on machine learning models of brain activity that predict sleep stage [3]. However, there is a distinction between using machine learning to find functional relationships and using machine learning to explain mechanistic properties [4]. For example, machine learning can effectively detect patterns of brain activity, but may not reveal a better understanding of the mechanisms that change brain function [5]. To address this limitation, this paper offers a synthesis of recent literature using a novel approach called *explainable artificial intelligence* that enables the extraction of information from machine learning models [6].

From a decision-making perspective, implementing deep brain stimulation requires a series of choices that artificial intelligence can optimize [7]. For example, closed-loop deep brain stimulation systems can use neural activity to detect symptoms and adapt stimulation parameters in real-time [8]. These systems could utilize computational models of brain activity that help decide how to alter stimulation and inhibit an unwanted behavior before it begins, such as a tremor [9]. This adaptive stimulation approach follows a sensor-trigger protocol that alters the stimulation parameters according to the difference between ongoing neural activity and a desired motif of neural activity. One potential way to create such a motif is to create a pre-trained computational model based on stimulation characteristics and a cost function that mirrors symptoms [10,11]. Moreover, animal models could improve understanding of how phasic and chronic deep brain stimulation differ in their efficacy to produce a specific neurotransmitter, such as dopamine [12]. The combination of different artificial intelligence implementations could serve as a clinical decision support system that improves deep brain stimulation by leveraging high-dimensional data using computational resources that exceed the abilities of human clinicians [13].

A key improvement needed for artificial intelligence algorithms in deep brain stimulation research is the ability to understand how the model made a decision. Extracting the logic for decisions made by machine learning algorithms is one way to improve the interpretability of artificial intelligence approaches to deep brain stimulation [14]. Extracting information from decision models can help clinicians understand the model’s characteristics and share this information with colleagues and patients [15]. For example, PECLIDES is a personalized clinical decision support system built using random forests to diagnose neurological diseases, such as Parkinson’s disease [16]. Besides making diagnoses, PECLIDES creates a set of if-then-rules that can explain to a physician how the model made the diagnosis.

One of the benefits of extracting explainable solutions from models for deep brain stimulation is maintaining human supervision of the algorithms [17]. For example, a study of clinical decision support systems used machine learning algorithms to build a model of deep brain stimulation parameter settings and medication dosages that predicted clinical outcomes following deep brain stimulation [18]. The model has the potential to show a physician the predicted clinical outcomes of different treatment strategies and suggest a specific treatment strategy that minimizes symptoms at follow-up [19]. However, the system does not extract rules from the decision trees, leaving the potential physician to wonder why the model predicted a particular treatment strategy would minimize symptoms at follow-up. An explainable artificial intelligence approach to clinical decision support systems could extract a set of rules explaining the decision, as well as an example of the critical values for a patient concerning factors heavily weighted in the decision [20].

The aim of this paper is to synthesize recent literature on explainable artificial intelligence approaches to deep brain stimulation [21]. There is evidence that machine learning models can predict treatment outcomes and identify treatment targets [22]. It has also been reported that in the domain of closed-loop brain stimulation, explainable artificial intelligence can improve treatment outcomes and advance fundamental knowledge about brain-stimulation relationships [23]. However, explainable artificial intelligence is an emerging field that is not integrated into deep brain stimulation research [24]. Here, a topic modeling approach is used to synthesize themes and trends in the literature regarding approaches to extracting domain knowledge from machine learning models relevant to deep brain stimulation. Topic modeling is an algorithmic approach to discovering themes in a corpus of text [25,26]. The impetus of this project was to synthesize the progress made in different aspects of deep brain stimulation, including the initial screening of patients for eligibility, planning for surgical implantation, and post-op follow-up. Thus, this paper offers a review at the intersection of explainable artificial intelligence and deep brain stimulation across the treatment sequence.

## 2. Topic Modeling Procedure

### 2.1. Overview

To guide the review, a topic modeling algorithm (Latent Dirichlet Allocation) was used to discover themes and trends in the applicable journal articles. Latent Dirichlet Allocation is an unsupervised learning method based on a generative probabilistic model [27]. In short, Latent Dirichlet Allocation finds latent thematic structures by detecting a specified number of topics through analysis of the probabilities of words in a collection of documents. The methods described below were pre-registered on the open science foundation using the PRISMA 2020 checklist (https://osf.io/vdx6s/?view_only=752fbfdce1ea4ffbb360e3d09f22fdf3, accessed on 14 December 2022) [28]. The code used for all analyses is available in the Appendix A.

### 2.2. Search for Journal Articles

The first step in identifying topics at the intersection of explainable artificial intelligence and deep brain stimulation is curating relevant journal articles. Thesearch was conducted using https://scholar.google.com/ (accessed 4 January 2023) with the following terms: (“deep brain stimulation” “explainable artificial intelligence” OR “interpretable artificial intelligence” OR “interpretable machine learning”). This search resulted in 79 articles.

### 2.3. Inclusion and Exclusion Criteria

The first inclusion criterion was that the journal articles matched the keyword search. The google scholar results were screened by the lead author (BA) and excluded if they were book chapters (n = 17), conference proceedings (n = 8), lists of articles (n = 1), dissertations (n = 3), duplicates (n = 7), or webpages (n = 1).

### 2.4. Text Pre-Processing

The remaining 40 articles were converted to text files, reference sections were removed, and the remaining text was examined for errors using spell check through Google docs. The remaining preprocessing and analysis was conducted using the R programming language (version 4.2.1, 23 June 2022). Trigrams (i.e., three consecutive words) were extracted from each journal article and inspected to filter out articles that only contained search terms in the reference section (n = 28). The resulting 12 articles were published between 2018 and 2022 and directly related to deep brain stimulation and explainable or interpretable artificial intelligence (see Table 1).

Next, paragraphs from each journal article were segmented into separate files (n = 1262). The R package tm (version 0.7-8) was used to merge all paragraphs into a corpus, remove stopwords, symbols, numbers, and punctuation, and to lemmatize each word. The package tm was also used to tokenize the text into trigrams for keyword screening and bigrams for final analysis. Following the initial computation of the final topic model, additional irrelevant bigrams were removed from the corpus and the topic models were estimated again (i.e., may_also, studi_us, sourcerightclick_figur, model_, et_al, use_data, sampl_siz, studi_focus, al_b, can_provid, updr_iii, s_disea, gof_valu, can_us, patient_, paper_entitl, use_machin, fig_b, can_us, et_al, paper_entitl, use_data, input_data, studi_us, n_n, n_rem, data_may, s_disea, data_set, model_, can_also, wide_us, g_figur, file_, al_b).

### 2.5. Topic Modeling

One requirement for a topic model is the specification of the number of topics. This was accomplished using two approaches. First, the R package *ldatuning* (version 1.0.2) was used to find the optimal number of topics based on metrics of coherence for each topic. To ensure that any one set of paragraphs did not overly influence this process, the paragraphs were split into ten random subsets, and four different coherence metrics were computed for topic models with 2 to 20 topics. These coherence metrics were estimated ten times, each time removing one of the paragraph subsets. The median coherence scores across the ten iterations were computed for each metric and each topic model. Two of the coherence metrics show more coherent topics with lower values (i.e., Arun2010, CaoJuan2009), whereas the other two metrics show more coherent topics with higher values (i.e., Griffiths2004, Deveaud2014). The average intersection of these lines was computed, which suggested the optimal number of topics was nine, based on coherence.

The second approach to finding the optimal number of topics was based on the perplexity score of models with different numbers of topics using the LDA function from the R package *topicmodels* (version 0.2-12). Perplexity is a metric that shows how probable a collection of text is, given a topic model. For this process, a topic model was estimated for 2 to 20 topics using nine of the ten subsets of paragraphs, and that topic model was then applied to the remaining fold where perplexity was computed. This process was repeated until each fold was used as the holdout sample for cross-validation. The median perplexity across all folds was computed for each number of topics. The perplexity measure was rescaled to a range of 0 to 1, and the optimal number of topics was computed based on the number of topics when the perplexity measure decreased less than 1%. This process resulted in the optimal topic number being 11, based on perplexity.

Given the similar number of topics determined using coherence and perplexity, the smaller number (nine topics) was then used to estimate the ultimate model based on the entire corpus. The Latent Dirichlet Allocation model was estimated by applying the Gibbs algorithm for its convergence method.

## 3. Topics in Deep Brain Stimulation Research Using Explainable Artificial Intelligence

A Latent Dirichlet Allocation model was used to estimate nine topics that summarized the journal articles matching the search terms. The ultimate model was interpreted based on the ten most probable bigrams per topic, and the ten most probable paragraphs per topic. For interpretation, each of the top ten paragraphs per topic were traced back to their original paper and described in the results regarding deep brain stimulation and explainable or interpretable artificial intelligence. After interpreting each of the nine topics, they were grouped together based on similar themes: patient classification, precision psychiatry, complex systems, methodological concerns, heterogeneity of treatment response, automated symptom assessment, and complex systems (see Table 2 for themes and top five topic bigrams).

### 3.1. Patient Classification

The first topic in the subject area of patient classification relates to the field of precision psychiatry, an area in which machine learning can improve the diagnosis, prognosis, and treatment [29,39]. For instance, Drysdale et al. [40] used unsupervised machine learning methods of clustering to discover subtypes of patients with depression based on neuroimaging data. These subtypes showed differential response rates to brain stimulation, suggesting machine learning can help identify patients who are more or less likely to respond to brain stimulation. Importantly, this study also used an interpretable machine learning model for prediction (i.e., support-vector machine) and identified the most important neuroimaging features that discriminated between brain stimulation responders vs. non-responders. Such information is important because it advances understanding of how heterogeneous symptom profiles in brain stimulation patients relate to patterns of brain activity. It also represents an example of using interpretable machine learning models and the explicit extraction of domain knowledge about neurobiological heterogeneity and the efficacy of brain stimulation.

The study by Drysdale et al. [40] paints the picture of using machine learning models of behavior and neurophysiology to inform medical treatment with deep brain stimulation [29]. It may also be important to adapt stimulation strategies based on momentary changes in behavior and neurophysiology within a person across time. For example, an adaptive, closed-loop stimulation system is being investigated in mice [41]. One interesting component of this system is that it doesn’t use a pre-trained model of neural activity but learns the stimulation protocol individually based on the difference between the evoked response and the desired target response. The system iteratively updates stimulation parameters until it minimizes this difference. With a combination of these types of adaptive brain stimulation approaches with interpretable models, studying the final parameters across a group of people could help to better understand how different stimulation strategies achieve desired brain states or eliminate unwanted behaviors.

The second topic in the subject area of patient classification relates to using interpretable machine learning models of behavioral and neurophysiological data to classify patients into broad diagnostic categories. For example, Jung et al. [32] used regularized logistic regression to discriminate patients with Parkinson’s disease from healthy controls based on features from functional neuroimaging scans. However, even interpretable models that are algorithmically simple still require post-hoc explainability methods to communicate to clinicians how the model works. Pinto [34] suggests that artificial intelligence applications which include machine learning models of electroencephalogram data to make diagnostic decisions, should also provide an annotated electroencephalogram recording that visualizes important features for the decision. A potential application for neuroimaging data could be to visualize functional neuroimaging data in brain regions important for diagnostic decisions.

### 3.2. Precision Psychiatry

The first topic in the subject area of precision psychiatry concerns using machine learning models to discover within-person differences relevant to the efficacy of deep brain stimulation. For example, Chen et al. [3] investigated support vector and decision tree models that classified sleep-stages based on local field potentials recorded from the stimulation electrode. The authors extracted the importance of different frequencies of local field potentials for sleep stage classification and identified beta, alpha, and gamma rhythms in the subthalamic nucleus as the best predictors of sleep stage. Such an interpretation could be critical for the development of closed-loop deep brain stimulation systems that adapt stimulation to the sleep-stage. Another approach to optimizing deep brain stimulation is using treatment response prediction models based on changes in local field potentials following intraoperative brain stimulation [36]. This paper follows a similar feature selection approach with an interpretable logistic regression model to extract electrophysiological biomarkers of treatment effects. Alternatively, it may be possible to identify treatment response biomarkers using a classification model based on neuroimaging data [32].

The second topic in the subject area of precision psychiatry concerns the use of explainable methods to identify between-person differences that can influence the efficacy of deep brain stimulation. For example, Habets et al. [30] used an interpretable logistic regression prediction model of preoperative clinical variables to predict clinically relevant improvement in patients who received deep brain stimulation. Using the predictive weight of each variable in the model to identify important predictors, the authors discovered younger Parkinson’s disease patients were more likely to respond to deep brain stimulation. However, one of the most common problems with clustering patients into discrete groups is the lack of using a rigorous approach for selecting the number of clusters [31]. Anatomical and pathophysiological differences between patients may not result in coherent groups at all, resulting in inconsistent clinical outcomes [38,42].

### 3.3. Methodological Concerns

The first topic in the subject area of methodological concerns is the importance of explainability over interpretable models when using artificial intelligence to make high-risk medical decisions, such as the prescription of deep brain stimulation [34]. While an interpretable model can facilitate explainability, there may still be a need to apply explainable methods post hoc. Using explainable methods can keep humans in the loop by identifying what features of a model caused a particular decision, giving the model creators an opportunity to identify bias and update the decision-making process as needed [23,43]. Another benefit of being able to explain a decision made by an artificial intelligence system is that providers and patients are more likely to trust the decision if they can understand it. Importantly, there is often a difference in the accuracy of interpretable models with a minimal number of features compared to more complex models that are difficult to interpret, such as deep learning models [44]. Thus, addressing the social barriers to using artificial intelligence in deep brain stimulation may include applying post-hoc explainability methods to the most accurate models to explain to the patient and physician why a particular decision was made.

The second topic in the subject area of methodological concerns is about best practices and pitfalls when using machine learning [31]. For example, the curse of dimensionality is a situation in which a model has many features and few observations. Such a problem is common with neuroimaging data, where there are many measurements from the brain, yet only a few participants in the study, resulting in over-fitting. Concerning model complexity, there is a bias-variance trade-off. Simple models are high in bias (i.e., incorrect model assumptions) but are low in variance (i.e., sensitivity to noise), whereas complex models are low in bias but high in variance. The goal is to find an optimal level of model complexity that minimizes both bias and variance and results in a generalizable model. Finally, a common problem in machine learning studies of clinical populations is the small number of observations. A recent review of studies using machine learning methods in human movement biomechanics research showed that the median sample size was 40 [31]. Sample size is a concern because it is necessary to split the sample into training and validation subsets before performing feature engineering, hyperparameter tuning, and model validation.

### 3.4. Complex Systems

The theme of using machine learning to model complex systems consisted of one topic. While machine learning is often used for prediction, it can also model dynamic brain activity following stimulation [37]. This is important because deep brain stimulation can be considered an attempt to control and optimize the topology of a complex network of brain regions. In a simulation study, Krylov et al. [45] used a reinforcement learning paradigm to suppress unwanted synchronous oscillations in degenerative neurons to treat Parkinson’s disease. Such an application of reinforcement learning could create individual models specific to each patient’s brain structure. While the paper by Krylov et al. [45] did not include explainable methods, there are ways to extract a summary of the policy the reinforcement learning agent has learned [46]. The extraction of these policies could provide insight on how to stimulate the brain optimally.

### 3.5. Automated Symptom Assessment

The theme of assessing patient symptoms automatically using machine learning consisted of one topic. Specifically, the topic encompassed the possibility of using explainable artificial intelligence to develop prediction models of Parkinson’s disease symptoms based on videos of patient movement [35]. As motor function is a key impairment of movement disorders, the authors aimed to develop an automated gait assessment. After the prediction model was developed, the authors extracted Shapley additive explanations values for each feature in the model, which provides a metric of the importance for each model feature in making a prediction. Shapley additive explanations values have the potential to explain to clinicians why a patient’s symptom severity is rated at a particular level. This type of automated assessment could be integrated with deep brain stimulation in a way that titrates stimulation based on symptom severity.

### 3.6. Heterogeneity of Treatment Response

The theme of using machine learning to predict treatment response comprised only one topic. At the heart of this topic is the problem of heterogeneity within and co-morbidity between diagnostic groups. The goal is to use models of symptoms, demographics, and neurobiology to account for this heterogeneity and make more precise treatment recommendations. One potential solution is using latent space-based supervised learning to uncover latent dimensions of neural circuits in psychiatric disorders [29]. For example, Wu et al. [47], used latent space-based supervised learning to discover neurobiological signatures that predicted response to antidepressant treatment. Similar signatures likely exist that account for heterogeneity in response to deep brain stimulation. An explainability approach called accumulated-local-effects could be applied to machine learning models to extract information about how different levels of a feature are related to treatment response [33,48]. Accumulated-local-effects may be helpful because they provide information about the directionality of effects, in addition to the importance of the feature in making a prediction.

## 4. Discussion

This paper presents a review of topics and themes in deep brain stimulation research using explainable artificial intelligence. The resulting topics were derived using topic modeling of the full text of 12 journal articles that matched a keyword google scholar search (“deep brain stimulation” AND (“explainable artificial intelligence” OR “interpretable machine learning” OR “interpretable artificial intelligence”). The results show themes that include patient classification, precision psychiatry, complex systems, methodological concerns, heterogeneity of treatment response, automated symptom assessment, and complex systems.

The results show a systematic evaluation of current directions in deep brain stimulation research using explainable artificial intelligence, but also have the potential to highlight underserved areas of research. For example, many of the topics and studies reviewed concern patient classification and optimizing the identification of patients who will or won’t benefit from deep brain stimulation. Conversely, only one study investigated models that can optimize a brain stimulation device [45]. This study used a reinforcement learning method but did not use explainable methods and only mentioned them in passing. However, there are methods of extracting meaning from reinforcement learning models [46], suggesting that applying explainable methods to machine learning models of complex systems relevant to deep brain stimulation is an open research area. Such research has the potential to advance understanding of how stimulation protocols can minimize patient symptoms.

As explainable artificial intelligence is a new and burgeoning field, our results show a limited scope of explainable artificial intelligence in deep brain stimulation research. It is important to note that the scarcity of explainable methods is acting as a social barrier to implementing machine learning models in deep brain stimulation treatment [34]. Not only will explainable methods build trust by opening the black box to clinicians and patients, but they also have the potential to advance understanding of how, when, and why deep brain stimulation is effective. Thus, future deep brain stimulation researchers are encouraged to use explainable methods when examining the role of artificial intelligence. Journal editors and reviewers are also encouraged to advise authors to include explainability as a component of their studies. General sources about explainable methods [48] and use-cases from other domains [49] are likely sources of inspiration.

### Limitations and Future Directions

A limitation of the topics discovered is that the results in this paper are from a topic modeling analysis of papers explicitly mentioning explainable or interpretable artificial intelligence. Thus, this study doesn’t show themes from the broader literature on artificial intelligence and deep brain stimulation. Instead, this paper shows the themes that researchers have investigated and emphasizes opportunities for future research. Excellent reviews exist on implementing machine learning models in deep brain stimulation research [22], and the studies reported provide examples of future research projects that could incorporate explainable methods.

A second limitation is that the journal articles analyzed in this paper were not all open-source, making the data not publicly shareable. However, this report includes a list of analyzed articles. The code is available, though the reader should know that topic modeling is not the most sophisticated language model available. Future iterations of language models about deep brain stimulation could use a more advanced approach, such as a generative pre-trained transformer model validated by experts in the field [50].

Explaining machine learning models of neural activity following different stimulation parameters could reveal key mechanisms for theoretical models of deep brain stimulation and improve treatment efficacy [51]. Extracting feature importance from a model of stimulation parameters could help identify a biological subspace for potential mechanisms of change [23]. Explainable artificial intelligence approaches to studies of stimulation patterns could advance the theory of how levels of tonic and phasic neural activity impact neurotransmitter release [12]. Personalized medicine is a translational application of explainable artificial intelligence [52,53]. For example, the development of a closed-loop deep brain stimulation system that automatically adjusts stimulation parameters for each individual patient [54]. Or a warning system that predicts adverse effects and complications caused by stimulation [55]. Together, these theoretical and translational applications will enhance the efficacy and reliability of deep brain stimulation.

Researchers at the intersection of explainable artificial intelligence and deep brain stimulation need to work towards improving answers to both the methodological and theoretical questions. From a methodological perspective, a tremendous obstacle to building machine learning models is the small amount of publicly available data. Similar to the ABCD study, the field needs a multi-site, publicly available, nationally representative dataset from deep brain stimulation patients [56]. An open question is whether personalized systems need a smaller amount of patient data to build a trustworthy model. From a theoretical perspective, explainable artificial intelligence approaches would benefit from integrating computational models of brain dynamics [57,58]. In the end, these efforts may lead to an adaptive system governed by theoretical models of brain activity.

## 5. Conclusions

This paper offers a current understanding of themes surrounding explainable artificial intelligence and deep brain stimulation research. Explainable artificial intelligence has the potential to make theoretical and translational contributions to deep brain stimulation research. From a theoretical perspective, explainable artificial intelligence has the potential to improve understanding of neural mechanisms. For example, visualizing brain regions activated by different parameter settings could improve understanding of how brain circuits respond to stimulation. From a translational perspective, explainable artificial intelligence has the potential to improve individual specific stimulation strategies that improve patient outcomes. Overall, the intersection of these two fields is in its infancy and new themes will emerge once there are more implementations reported in the literature.

## Figures and Tables

**Table 1 biomedicines-11-00771-t001:** Final corpus of journal articles.

Authors	Year	Title	Journal
Chen, Y., Gong, C., Hao, H., Guo, Y., Xu, S., Zhang, Y., … and Li, L.	2019	Automatic sleep stage classification based on subthalamic local field potentials [3]	IEEE Transactions on Neural Systems and Rehabilitation Engineering
Chen, Z. S., Galatzer-Levy, I. R., Bigio, B., Nasca, C., and Zhang, Y.	2022	Modern views of machine learning for precision psychiatry [29]	Patterns
Fellous, J. M., Sapiro, G., Rossi, A., Mayberg, H., and Ferrante, M.	2019	Explainable artificial intelligence for neuroscience: behavioral neurostimulation [23]	Frontiers in neuroscience
Habets, J. G., Janssen, M. L., Duits, A. A., Sijben, L. C., Mulders, A. E., De Greef, B., … and Herff, C.	2020	Machine learning prediction of motor response after deep brain stimulation in Parkinson’s disease—proof of principle in a retrospective cohort [30]	PeerJ
Halilaj, E., Rajagopal, A., Fiterau, M., Hicks, J. L., Hastie, T. J., and Delp, S. L.	2018	Machine learning in human movement biomechanics: Best practices, common pitfalls, and new opportunities [31]	Journal of biomechanics
Jung, K., Florin, E., Patil, K. R., Caspers, J., Rubbert, C., Eickhoff, S. B., and Popovych, O. V.	2023	Whole-brain dynamical modelling for classification of Parkinson’s disease [32]	Brain Communications
Padberg, F., Bulubas, L., Mizutani-Tiebel, Y., Burkhardt, G., Kranz, G. S., Koutsouleris, N., … and Brunoni, A. R.	2021	The intervention, the patient and the illness–personalizing non-invasive brain stimulation in psychiatry [33]	Experimental Neurology
Pinto, M. F., Leal, A., Lopes, F., Pais, J., Dourado, A., Sales, F., … and Teixeira, C. A.	2022	On the clinical acceptance of black-box systems for EEG seizure prediction [34]	Epilepsia Open
Rupprechter, S., Morinan, G., Peng, Y., Foltynie, T., Sibley, K., Weil, R. S., … and O’Keeffe, J.	2021	A clinically interpretable computer-vision based method for quantifying gait in Parkinson’s disease [35]	Sensors
Sendi, M. S., Waters, A. C., Tiruvadi, V., Riva-Posse, P., Crowell, A., Isbaine, F., … and Mahmoudi, B.	2021	Intraoperative neural signals predict rapid antidepressant effects of deep brain stimulation [36]	Translational psychiatry
Tang, Y., Kurths, J., Lin, W., Ott, E., and Kocarev, L.	2020	Introduction to focus issue: When machine learning meets complex systems: Networks, chaos, and nonlinear dynamics [37]	Chaos: An Interdisciplinary Journal of Nonlinear Science
Zdravkova, K., Krasniqi, V., Dalipi, F., and Ferati, M.	2022	Cutting-edge communication and learning assistive technologies for disabled children: An artificial intelligence perspective [38]	Frontiers in Artificial Intelligence

**Table 2 biomedicines-11-00771-t002:** Themes and Top Five Topic Terms.

Topic	Theme	Top 5 Bigrams
1	Patient Classification	mental health, deep learning, assistive technology, precision psychiatry, mental disorder
2		model fit, Parkinson disease, model parameter, filter condition, behavior model
3	Precision Psychiatry	sleep stage, beta power, closed loop deep brain stimulation, vote length, wake sleep
4		prediction model, Unified Parkinson’s Disease Rating Scale, weak responder, subthalamic nucleus deep brain stimulation, model can
5	Methodological Concerns	seizure prediction, clinical trial, support vector machine model, previous studies, decision tree model
6		neural network, feature selection, model performance, test set, support vector machine model
7	Complex Systems	machine learning, reservoir computing, learning method, complex system, dynamic system
8	Automated Symptom Assessment	feature value, step frequency, Parkinson’s disease patient, model estimation, arm swing
9	Heterogeneity of Treatment Response	psychiatric disorder, brain stimulation, functional connectivity, deep brain, neural circuit

To improve interpretability, the lemmatized bigrams are reverted into their original meaning. Bigrams are separated by commas, and some bigrams may include more than two terms because of acronyms, such as PD for Parkinson’s disease.

## Data Availability

The data used in this report comes from published articles that are copyrighted. However, the list of articles used are listed in this report and the code used to analyze them is provided in Appendix A.

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
