# Peer review of "Discovering Themes in Deep Brain Stimulation Research Using Explainable Artificial Intelligence"

_biomedicines, 2023, doi:10.3390/biomedicines11030771_

Round 1

Reviewer 1 Report

Dear Author,

I appreciate your thorough analysis of the field of explainable ai in deep brain stimulation. I too believe that AI will bring a level of treatment personalization that is unatainable in its absence. The filed is still thin and you mention that. More studies and consequent reviews are needed to talk about a mature topic.

I have only one comment and it relates to table 2 which represents directly the findings. However, many of the abbreviations used are not common and I think they would benefit from explanatory paranthessis (e.g. updr_iii, or stn_dbs)

Author Response

Thanks to the reviewer for their comment on abbreviations and explaining bigrams in the table. Table 2 now has interpretable bigrams, to help the reader understand the top terms of each topic. I exchanged the lemmatized terms to interpretable terms based on the source articles. Now, instead of 'precis_psychiatri', the table reads 'precision psychiatry'.

Reviewer 2 Report

Allen in the present review article entitled ‘Discovering themes in deep brain stimulation research using explainable artificial intelligence’, investigated the current status of knowledge of artificial intelligence approaches to deep brain stimulation. For this purpose, the author used a topic modelling procedure to synthesize themes and trends in the literature regarding approaches to extracting domain knowledge from machine learning models relevant to deep brain stimulation. Results showed a limited scope of explainable artificial intelligence in deep brain stimulation research, but that could be caused that the scarcity of explainable methods which acts as a social barrier to implementing machine learning models in deep brain stimulation treatment. Therefore, the author concluded by stating that future deep brain stimulation researchers are encouraged to use explainable methods when examining the role of artificial intelligence.

The main strength of this paper is that it addresses an interesting and timely question, providing an overview of the current concept of themes surrounding explainable artificial intelligence and deep brain stimulation research. In general, I think the idea of this manuscript is really interesting and the authors’ fascinating observations on this timely topic may be of interest to the readers of Biomedicines. However, some comments, as well as some crucial evidence that should be included to support the author’s argumentation, needed to be addressed to improve the quality of the manuscript, its adequacy, and its readability prior to the publication in the present form.

Please consider the following comments:

A graphical abstract that will visually summarize the main findings of the manuscript is highly recommended.

Abstract: In my opinion, a lack of explanation of machine learning definition and how it mostly can be successfully used in a wide range of healthcare problems, including deep brain stimulation, the reader unable to grasp the key concept of this article by looking at the abstract. I would suggest to reorganize this section, making sure to include a detailed explanation of application of machine learning in healthcare, making sure to highlight the potential benefits of machine learning in treatment with deep brain stimulation. 

In general, I recommend Author to use more references to back his claims, especially in the Introduction of this review, which I believe is lacking. Thus, I recommend the author to attempt to expand the topic of this manuscript, as the bibliography is too concise. Nevertheless, I believe that less than 50 articles are too low for a Review article. Therefore, I suggest focus on researching relevant literature: in my opinion, adding more citations will help to provide better and more accurate background to this study. 

I would ask the Author to clarify the criteria they decided to use for studies’ collection in their review: he should specify the requirements used to decide whether a study met the inclusion/exclusion criteria of the review, describe whether he included a balanced coverage of all information that is actually available, whether he has included the most recent and relevant studies and enough material to show the development and limitations in this field of interest. Finally, I believe that he should briefly present results of all statistical syntheses conducted.

The objectives of this study are generally clear and to the point; however, I believe that there are some ambiguous points that require clarification or refining. In my opinion, author should be explicit regarding how he wanted to assess the applications of machine learning approaches for deep brain stimulation.

Discussion: In this final section, Author described the results and their argumentation and captured the state of the art well; however, I would have liked to see some views on a way forward. I believe that the author should make an effort, trying to explain the theoretical implication as well as the translational application of this research article, to adequately convey what he believes is the take-home message of this study. Discussion of theoretical and methodological avenues in need of refinement is necessary, as well as suggestions of a path forward in understanding how application of artificial intelligence in deep brain stimulation could enable the interpretation of large multimodal datasets and help to inform DBS candidate selection, surgical targeting, programming optimization, and DBS mechanisms, potentially paving the way for precision neuromodulation. In this regard, In this regard, recent evidence suggests that the application of new methods in neurodegenerative disorders treatment, such as the Non-invasive brain stimulation techniques (NIBS), have shown promising results in humans (doi: 10.1038/s41598-019-51164-2). Importantly, I recommend referring recent studies that revealed that the application of NIBS induces long-lasting effects, noninvasively modulating the cortical excitability, and modulates a variety of cognitive functions, like memory and emotional learning (https://doi.org/10.3389/fnbeh.2022.998714).

In my opinion, I think the ‘Conclusions’ paragraph would benefit from some thoughtful as well as in-depth considerations by the Author, because as it stands, it is very descriptive but not enough theoretical as a discussion should be. Author should make an effort, trying to explain the theoretical implication as well as the translational application of his research.

I would ask the Author to include a proper and defined ‘Limitations and future directions’ section before the end of the manuscript, in which he can describe in detail and report all the technical issues that could be brought to the surface.

Table: According to the Journal’s guidelines, please provide a short explanatory caption for the table within the text.

References: Author should consider revising the bibliography, as there are several incorrect citations. Indeed, according to the Journal’s guidelines, they should provide the abbreviated journal name in italics, the year of publication in bold, the volume number in italics for all the references.

I hope that, after these careful revisions, the manuscript can meet the Journal’s high standards for publication. I am available for a new round of revision of this article. 

Best regards,

Reviewer

Author Response

Thank you to the reviewer for their comments. The graphical abstract is revised and now provides a visual summary of the themes discovered in the results. The abstract is revised and now includes more information on machine learning and highlights the potential benefits of machine learning in treatment with deep brain stimulation. The introduction is now expanded and the paper now has more than 50 references. The introduction now has a better offering of background information relevant to the paper. Inclusion and Exclusion Criteria are now explicitly labeled and described. The topics described span across all the included journal articles. The introduction now includes an explicit statement about the initial plan regarding the synthesis of the different applications. There is now a future directions section that discusses theoretical and methodological avenues. The conclusions paragraph now highlights theoretical and translational applications. The paper now includes limitations and future directions sections. All tables have a short explanatory note in the text. The citations are now formatted according to the journal specifications. 

Reviewer 3 Report

The Manuscript: „ Discovering themes in deep brain stimulation research using explainable artificial intelligence’’ by Ben Allen synthesized recent literature on explainable artificial intelligence approaches to deep brain stimulation. The review amply offers a current understanding of themes surrounding explainable artificial intelligence and deep brain stimulation research. After going through the manuscript, I have a couple of comments for the author:

1.     Did EA treatment also alleviate the microglial activation in the hippocampus?

2.     It is known that Explainable Artificial Intelligence applied to the domain of closed-loop neurostimulation yields important new insights both at the fundamental research level and at the clinical therapeutic level. Please briefly discuss this point in the manuscript.

Author Response

The manuscript now mentions at the end of the introduction that It is known that Explainable Artificial Intelligence applied to the domain of closed-loop neurostimulation yields important new insights both at the fundamental research level and at the clinical therapeutic level. 

For the second comment, the paper does not mention microglial activation in the hippocampus, so it isn't clear what this comment is about.

Round 2

Reviewer 2 Report

The author did an excellent job clarifying all the questions I have raised in my previous round of review. Currently, this paper entitled ‘DISCOVERING THEMES IN DEEP BRAIN STIMULATION RESEARCH USING EXPLAINABLE ARTIFICIAL INTELLIGENCE’, is a well-written, timely piece of research that described artificial intelligence approaches to deep brain stimulation. 

Overall, this is a timely and needed work. It is well researched and nicely written, therefore I believe that this paper does not need a further revision, therefore the manuscript meets the Journal’s high standards for publication.

I am always available for other reviews of such interesting and important articles.

Thank You for your work, Reviewer